# Antiparasitic Constituents of *Beilschmiedia louisii* and *Beilschmiedia obscura* and Some Semisynthetic Derivatives (Lauraceae)

**DOI:** 10.3390/molecules25122862

**Published:** 2020-06-21

**Authors:** Christine C. Waleguele, Brice M. Mba’ning, Angelbert F. Awantu, Jean J. K. Bankeu, Yannick S. F. Fongang, Augustin S. Ngouela, Etienne Tsamo, Norbert Sewald, Bruno N. Lenta, Rui W. M. Krause

**Affiliations:** 1Department of Organic Chemistry, Faculty of Science, University of Yaoundé I, P.O. Box 812 Yaoundé, Cameroon; wwaleguele@yahoo.fr (C.C.W.); brice_mbaning@yahoo.fr (B.M.M.); sngouela@yahoo.fr (A.S.N.); tsamoet@yahoo.fr (E.T.); 2Department of Chemistry, Faculty of Science, The University of Bamenda, P.O. Box 39 Bambili, Cameroon; aawantu@gmail.com (A.F.A.); bankeu@gmail.com (J.J.K.B.); 3Department of Chemistry, Higher Teacher Training College, The University of Maroua, P.O. Box 55 Maroua, Cameroon; 4Department of Chemistry, Bielefeld University, P.O. Box 100131, 33501 Bielefeld, Germany; norbert.sewald@uni-bielefeld.de; 5Department of Chemistry, Higher Teacher Training College, University of Yaoundé I, P.O. Box 47 Yaoundé, Cameroon; 6Department of Chemistry, Rhodes University, P.O. Box 94, Grahamstown 6140, South Africa; r.krause@ru.ac.za

**Keywords:** Lauraceae, *Beilschmiedia obscura*, *Beilschmiedia louisii*, endiandric acid, antitrypanosomal activity

## Abstract

The MeOH/CH_2_Cl_2_ (1:1) extracts of the roots and leaves of *Beilschmiedia louisii* and *B. obscura* showed potent antitrypanosomal activity during preliminary screening on *Trypanosoma brucei brucei*. Phytochemical investigation of these extracts led to the isolation of a mixture of two new endiandric acid derivatives beilschmiedol B (**1**) and beilschmiedol C (**2**), and one new phenylalkene obscurene A (**3**) together with twelve known compounds (**4**–**15**). In addition, four new derivatives (**11a**–**11d**) were synthesized from compound **11**. Their structures were elucidated based on their NMR and MS data. Compounds **5**, **6**, and **7** were isolated for the first time from the *Beilschmiedia* genus. Additionally, the NMR data of compound **4** are given here for the first time. The isolates were evaluated for their antitrypanosomal and antimalarial activities against *Tb brucei* and the *Plasmodium falciparum* chloroquine-resistant strain *Pf3D7* in vitro, respectively. From the tested compounds, the mixture of new compounds **1** and **2** exhibited the most potent antitrypanosomal activity in vitro with IC_50_ value of 4.91 μM.

## 1. Introduction

The genus *Beilschmiedia* is one of the most widespread genera of the Lauraceae family. It is commonly distributed in tropical Asia and Africa and comprises of about 250 species. Forty-one of these species can be found in Cameroon, principally, in the South Western, Center; Eastern, Littoral, and Western regions [1,2]. Plants of the *Beilschmiedia* genus are largely used in traditional medicine in developing countries for the treatment of many ailments. The bark and leaves of some *Beilschmiedia* species have been reported to be used in traditional medicine for the treatment of uterine tumors, rubella, female genital infections, rheumatism, colon and digestive disorders, malaria, and a headache, as well as bacterial or fungal infections [3]. In Cameroon, the decoction of the bark of *B. obscura* is used in the center region for the treatment of microbial and parasitic infections [4]. The fruits of *B. louisii* are used as spice [5,6]. Previous phytochemical studies of plants of *Beileschmiedia* genus revealed the presence of various classes of secondary metabolites including alkaloids, arylpropanoids, benzopyrans, endiandric acids, and flavonoids [7]. Some of these phytoconstituents exhibited antibacterial, anticancer, antifungal, anti-inflammatory, antileishmanial, antiplasmodial, and cytotoxic activities as well as the *α*-glucosidase inhibitory activity during bioassay investigations [3]. To the best of our knowledge, no phytochemical and pharmacological studies have been reported on *B. louissi*. However, previous investigation of *B. obscura* reported the presence of one alkaloid, obscurine, and terpenoids [8]. Despite this phytochemical study, no attempt has been made so far to evaluate its pharmacological properties. In a continuing search for antimalarial and antitrypanosomal active compounds from Cameroonian medicinal plants, we have investigated the MeOH/CH_2_Cl_2_ (1:1) extracts of the stem bark of *B. obscura* and the roots of *B. louisii*, which showed antitrypanosomal activity during preliminary screening. Therefore, the purpose of the present study was to assess the antitrypanosomal and antiplasmodial activities of the extracts as well as those of the isolated compounds from these extracts. We report herein the isolation and structure elucidation of fifteen compounds from the roots of *B. louisii* and the stem bark of *B. obscura* as well as the structure modification of compound **11**, their antitrypanosomal, and antiplasmodial activities.

## 2. Results and Discussion

### 2.1. Structure Elucidation

The neutral fraction of the MeOH/CH_2_Cl_2_ (1:1) extract of stem bark of *B. obscura* was subjected to column chromatography (CC) over silica gel to afford beilschmiedol B (**1**), beilschmiedol C (**2**), and obscurene A (**3**). In addition, four known compounds including pentacosan-1-ene (**4**) [9,10], 15-phenylpentadecanoic acid (**5**) [11], (3*S*,4*R*,5*S*)-4-hydroxy-5-methyl-3-(11′-phenyl-1′-*n*-undecyl)-butanolide (**6**), and (3*S*,4*R*,5*S*)-4-hydroxy-5-methyl-3-(13′-phenyl-1′-*n*-tridecyl)-butanolide (**7**) [12] were isolated.

The neutral fraction of the MeOH/CH_2_Cl_2_ (1:1) extract of roots of *B. louisii* was subjected to CC over silica gel to afford eight known compounds including beilschmiedic acid A (**8**), beilschmiedic acid C (**9**), beilschmiedic acid D (**10**), beilschmiedic acid E (**11**) [13,14], methylenedioxyendiandric acid A (**12**) [15], *β*-sitosterol (**13**) [16], epicatechin (**14**) [17], and *β*-sitosterol-3-*O*-*β*-D-glucopyranoside (**15**) [18]. Furthermore, four new derivatives (**11a**, **11b**, **11c**, and **11d**) were prepared from compound **11**. The structures of isolates and synthetic derivatives were determined using their spectroscopic data.

The reduction of beilschmiedic acid E (**11**) [13] using lithium aluminum hydride (LiAlH_4_) [19] led to compound **11a** (Scheme 1) as an off-yellow oil. Its ^13^C–NMR spectrum (Table 1; Appendix A) showed 22 carbon signals, which were sorted by distortionless enhancement by polarization transfer (DEPT; Appendix A) and heteronuclear single quantum correlation (HSQC; Appendix A) experiments into twelve methine carbon resonances (including four olefinic carbons at *δ*_C_ 127.9 (C-5), 128.8 (C-9), 131.4 (C-8), and 133.9 (C-4)), nine methylene carbon signals (including one oxymethylene at *δ*_C_ 67.3 (C-1″)), and one methyl carbon signal at *δ*_C_ 14.3 (C-8′). In addition, the ^13^C–NMR spectrum of **11a** exhibited characteristic carbon signals for a tetracyclic alcohol with an endiandric acid skeleton containing the Δ^4,5^ and Δ^8,9^ carbon–carbon double bond equivalents [4,13,14,15,20,21]. The ^1^H–NMR spectrum (Table 1; Appendix A) showed signals of four olefinic protons at *δ*_H_ 6.10 (1H, dt, *J* = 10.0, 2.4 Hz, H-4), 5.50 (1H, dt, *J* = 10.0, 2.8 Hz, H-5), 5.39 (1H, brdd, *J* = 10.0, 0.8 Hz, H-8), and 5.56 (1H, dt, *J* = 10.0, 3.2 Hz, H-9) and a multiplet of an oxymethylene at *δ*_H_ 3.56 (2H, m, H-1″). Furthermore, the ^1^H–NMR spectrum showed signals of a methyl at *δ*_H_ 0.85 (3H, t, *J* = 6.8 Hz, H-8′), and seven methylene groups at *δ*_H_ 2.55–1.23 (14H, brs, H-1′ to H-7′), characteristic of an *n*-octyl moiety [4,13,14,15,20,21]. The location of both the double bonds and the oxymethylene was confirmed from the HMBC spectrum (Appendix A) where correlations of H-4/C-3, C-6, and C-13, H-5/C-3, C-6, C-7, and C-1″, and H-8/C-6, C-7, and C-10, were observed. These correlations confirmed that compound **11a** is an Δ^4,5^ and Δ^8,9^ unsaturated tetracyclic endiandric acid derived alcohol with an oxymethylene at C-6. Based on these information compound **11a** was concluded to be the product of reduction of the carboxylic group of beilschmiedic acid E (**11**) to which a trivial name beilschmiedol A (Figure 1) was given.

The oxidation of the alcoholic group of **11a** using the Dess Martin periodinane as an oxidizing reagent [22] yielded the aldehyde **11b** (Scheme 1) as an off-yellow oil. In fact, the ^1^H and ^13^C–NMR data of **11b** (Table 2; Appendix A) were nearly similar to those of compound **11a**. However, some noticeable discrepancies were observed on the ^1^H–NMR spectrum where the oxymethylene protons at *δ*_H_ 3.56 (2H, m, H-1″) were replaced by an aldehyde proton at *δ*_H_ 9.46 (1H, s, H-1″). In addition, the proton spectrum of **11b** exhibited the resonances of only three olefinic proton at *δ*_H_ 5.40 (1H, m, H-8), 5.49 (1H, m, H-9), and 6.81 (1H, t, *J* = 5.2 Hz, H-5) suggesting that a rearrangement had occurred. This was confirmed by the characteristic signal of the aldehyde carbonyl in the ^13^C–NMR spectrum at *δ*_C_ 193.9 (C-1″), replacing the oxymethylene carbon at *δ*_C_ 67.3 (C-1″) in compound **11a**. In addition, this spectrum also exhibited resonances of olefinic carbons at *δ*_C_ 154.0 (C-6), 145.9 (C-5), 127.8 (C-8), and 126.1 (C-9). Based on these data, compound **11b** was concluded to be a new aldehyde derivative to which the trivial name beilschmiedal (Figure 1) was given.

The derivatization of compound **11b** with thiosemicarbazide reagent [23] led to compound **11c** (Scheme 1) as a white amorphous powder. The molecular formula of **11c**, C_23_H_35_N_3_S, with eight degrees of unsaturation, was deduced from its NMR data and the high-resolution electrospray ionization (HRESIMS; Appendix A), which showed the protonated molecular ion peak [M + H]^+^ at *m*/*z* 386.2658 (calcd for C_23_H_34_N_3_S, 386.2624). The ^1^H and ^13^C–NMR data of the two compounds **11c** and **11b** (Table 2; Appendix A) were close to each other except some remarkable discrepancies. The main discrepancy observed in the ^13^C–NMR spectrum of **11c** was the signal of a conjugated imine at *δ*_C_ 145.5 (C-1″) in replacement of the aldehyde carbonyl group at *δ*_C_ 193.9 (C-1″) in **11b**. In addition, the ^13^C–NMR spectrum showed the signal of the thiocarbonyl carbon resonance at 180.4 (C-2″), the shielded Δ^5,6^ carbon resonances at *δ*_C_ 139.7 (C-6) and 139.5 (C-5). Discrepancies were also observed on the ^1^H–NMR spectrum, where the aldehyde proton at *δ*_H_ 9.46 (1H, s, H-1″) in **11b** was replaced by four other proton signals at *δ*_H_ 12.43 (1H, s, NH-1), 9.63 (1H, s, NH_2_-1), 8.39 (1H, s, H-1″), and 8.01 (1H, s, NH_2_-2). Based on the above evidences, compound **11c** was determined as a new beilschmiedic acid (D) derivative to which the trivial name beilschmiecarbazone (Figure 1) was given.

The nucleophilic addition of compound **11a** with 3,4,5-trimethoxybenzoic acid [24] led to compound **11d** (Scheme 1) as a white amorphous powder. The ^1^H and ^13^C–NMR data of **11d** (Table 2) displayed some similarities with those of **11a** except that there are additional signals due to the addition of the trimethoxygalloyloxy moiety. In fact, the ^13^C–NMR spectrum of **11d** (Appendix A) showed, in addition to signals of **11a**, one signal of a carboxyl group at *δ*_C_ 166.3 (C-1‴), signals of a 3,4,5-trisubstituted benzene ring at *δ*_C_ 125.3 (C-2‴), 106.8 (C-3‴/7‴), 152.9 (C-4‴/6‴), 142.2 (C-5‴), and signals of three oxymethyl groups at *δ*_C_ 60.9 (5‴-OMe) and 56.2 (4‴/6‴-OMe). This was further confirmed by the ^1^H–NMR spectrum (Appendix A), which exhibited a singlet of two protons at *δ*_H_ 7.23 (2H, s, H-3‴/7‴) and two singlets of three methoxy groups at *δ*_H_ 3.83 (6H, s, 3‴/5‴-OMe) and 3.85 (3H, s, 4‴-OMe). Based on the above data, compound **11d** was confirmed as a new derivative of beilschmiedic acid E (**11**) trivially named beilschmiegallate (Figure 1).

The mixture of compounds **1** and **2** was obtained as a yellowish oil with the same R_f_ on the thin layer chromatography (TLC) in different systems. Its HRESIMS spectra (Appendix A) showed the potassium adducts peak [M + K]^+^ at *m*/*z* 367.2303 (calcd for C_23_H_37_KO, 367.2398). The combination of this data with those of the ^1^H and ^13^C–NMR spectra clearly indicated that the yellowish sample was the mixture of two isomers. In fact, the ^13^C–NMR spectrum (Appendix A) supported by DEPT and HSQC (Appendix A) data displayed 34 carbon signals including one quaternary olefinic carbon at *δ*_C_ 141.6 (C-6, **2**), twenty-two methine carbons (including seven olefinic methine carbons at *δ*_C_ 126.0 (C-5, **2**), 126.1 (C-8, **2**), 127.9 (C-5, **1**), 128.2 (C-9, **2**), 128.9 (C-9, **1**), 131.3 (C-8, **1**), and 134.2 (C-4, **1**)), twenty-one methylene carbons (including two oxymethylenes at *δ*_C_ 65.5 (C-1″, **2**) and 67.6 (C-1″, **1**)), and methyl carbon signal at *δ*_C_ 14.3. The ^13^C–NMR spectrum of **1** and **2** (Table 1) also exhibited characteristic signals for molecules containing an aliphatic long chain moiety at *δ*_C_ 29.9–29.5. In addition, the ^13^C–NMR spectrum of the mixture exhibited characteristic carbon signals for two tetracyclic endiandric acid skeletons where, one contains an Δ^4,5^ and Δ^8,9^ and the other an Δ^5,6^ and Δ^8,9^ carbon–carbon double bond equivalents [4,13,14,15,20,21]. These observations were confirmed from the fact that carbon resonances of one of the constituents (**1)** were nearly similar to those of compound **11a**. The main dissimilarity between these compounds was at the level of their masses, which differ by 14 amu. The ^1^H–NMR spectrum (Table 1; Appendix A) of the mixture showed signals of seven olefinic proton resonances between *δ*_H_ 5.42–6.10, confirming the presence of the two tetracyclic endiandric acid skeletons, one with a Δ^4,5^ and Δ^8,9^ and the other with a Δ^5,6^ and Δ^8,9^ [4]. Key HMBC correlations (Figure 2 and Appendix A) observed between the oxymethylene proton at *δ*_H_ 4.12 (**2**) and the carbons at *δ*_C_ 34.4 (C-7, **2**), 126.0 (C-5, **2**), and 141.6 (C-6, **2**) and also between the olefinic proton at *δ*_H_ 5.74 (H-5, **2**) and the carbons at *δ*_C_ 34.4 (C-7, **2**), 36.8 (C-3, **2**), and 65.5 (C-1″, **2**) were useful to locate the methylene group at C-4 in compound **2**. Based on these data, it was concluded that compounds **1** and **2** were two new Δ^4,5^ and Δ^8,9^ and Δ^5,6^ and Δ^8,9^ alcohols with endiandric acid skeletons, to which the trivial names beilschmiedol B and C (Figure 1) were given, respectively.

The stereochemistry of all the endiandric acid derivatives was deduced from their biosynthesis. They are products of electrocyclic ring closures of naturally occurring polyketides, resulting from both the shikimate and acetate pathways. Their biosynthesis from the polyketide contains two consecutive non-enzymatic electrocyclic reactions, followed by an intramolecular Diels-Alder reaction [4,20,21,25]. As a result of the whole reaction sequence, an open-chain compound is converted into a tetracyclic compound. The starting product contains a conjugated tetraene system, as well as a conjugated diene system. Thus, it already displays the π electron systems required for the three pericyclic reactions; they are the two electrocyclizations and the Diels-Alder reaction [4,20,21,25].

Compound **3** was isolated as colorless oil. The molecular formula C_16_H_24_ accounting for five double bond equivalents was assigned based on NMR data (Appendix A) and HRESIMS, which showed the sodium adduct peak [M + Na]^+^ at *m*/*z* 239.1540 (calcd 239.1770 for C_16_H_24_Na). The ^13^C–NMR spectrum of compound **3** (Table 3) exhibited signals of 16 carbons, which were sorted by DEPT and HSQC into one quaternary carbon at *δ*_C_ 142.9 (C-1′), seven olefinic methine groups at *δ*_C_ 125.6 (C-4′), 128.4 (C-2′/6′), 128.2 (C-3′/5′), 130.1 (C-5), and 130.2 (C-6), seven methylene groups between *δ*_C_ 22.6 and 36.0, and one methyl group at *δ*_C_ 14.1 (C-10). Its ^1^H–NMR spectrum displayed signals for a triplet of one methyl at *δ*_H_ 0.80 (3H, t, *J* = 6.4 Hz, H-10), methylene groups between *δ*_H_ 1.19 and 2.70, two olefinic protons at *δ*_H_ 5.28 (2H, t, H-5,6), and a monosubstituted aromatic ring between *δ*_H_ 7.10 and 7.20. From this information, it can be deduced that compound **3** is an *n*-alkene with a phenyl moiety. The C-5/C-6 position of the double bond was deduced from the HRESIMS, which showed an intense peak at *m*/*z* 173.0524 corresponding to the loss of a C_3_H_7_- fragment. The geometry of the double bond was assigned as *cis* based on the chemical shift of C-4 (27.3) and C-7 (25.7). It is known that the signals of the carbons next to a *trans* double bond appear at δ ≈ 32, while those of a cis double bond appear at δ ≈ 27 [26,27]. The identification of the carbon atoms next to the double was made through the HMBC correlations of H-5/C-4, H-6/C-7, H_2_-4/C-5, and H_2_-7/C-6. However, most of these data were close to those of the previously described synthetic compound, benzene-5*E*-decen-1-yl with clear difference at the level of carbons adjacent to the double bond, which confers the *Z* configuration to compound **3** [28]. Based on the above data, compound **3** was assigned as a new stereoisomer, (5*Z*) 1-phenyldec-5-ene, to which the trivial name obscurene (Figure 1) was given.

Pentacos-1-ene (**4**) was obtained as colorless oil. Its molecular formula was assigned as C_25_H_50_ (one degree of unsaturation) according to HRESIMS, which showed a deprotonated molecular ion peak [M − H]^+^ at *m*/*z* 349.1892. The broad band decoupled ^13^C–NMR spectrum (Table 4; Appendix A) of **4** displayed characteristic signals for an aliphatic compound (including a methyl group at *δ*_C_14.1 (C-25) and a long chain carbon signals between 29.4 and 29.7) with terminal double bond (including a methylene at *δ*_C_ 139.3 (C-1) and an olefinic methine at *δ*_C_ 114.1 (C-2)). The ^1^HNMR spectrum (Appendix A) showed signals of a terminal aliphatic alkene at *δ*_H_ 4.96 (1H, brd, *J* = 10.2 Hz, H-1a), 5.02 (1H, brd, *J* = 17.4 Hz, H-1b), and 5.84 (1H, m, H-2). This spectrum also exhibited signals of a long chain carbon between 1.29 and 2.07 ppm and of a methyl group at *δ*_H_ 0.91 (3H, t, *J* = 7.2 Hz, H-25). Therefore, pentacos-1-ene (**4**) previously identified in the essential oil from the flowers of *Chrysanthemum coronarium* and from the cork of the wine through gas chromatography-mass spectrometry (GCMS) [9,10] is isolated here for the first time and its NMR data reported.

### 2.2. In Vitro Antiplasmodial and Antitrypanosomal Activities

Extracts and isolated compounds from both plants were screened for their in vitro antiparasitic activity against the *Plasmodium falciparum* strain *Pf3*D7 and *Trypanosoma brucei brucei* (Table 5). Samples that reduced parasite viability to < 20% were considered for dose-response assay to determine the IC_50_ value. The MeOH/CH_2_Cl_2_ (1:1) extract from the roots of *B. louisii* exhibited very good antitrypanosomal activity against *Tb brucei* with an IC_50_ value of 4.6 µg/mL, while the extracts of stem bark of *B. obscura* and leaves of *B. louisii* exhibited moderate activities against the same strain with IC_50_ values of 21.6 and 13.8 μg/mL, respectively (Table 5).

The non-alkaloid fraction of the roots extract of *B. louissi* was moderately active, while the alkaloid rich fraction was non-active. Regarding the leaves, only the total extract showed a moderate activity. In both cases, the synergetic effect of the alkaloid and the non-alkaloid fractions could be evident.

Each extract was partitioned into alkaloid and neutral fractions. Alkaloid fractions and neutral fractions were also evaluated for their antitrypanosomal and antimalarial activities. The alkaloid fractions of the stem bark of *B. obscura* exhibited moderate antitrypanosomal activity with an IC_50_ value of 16.9 µg/mL. Amongst the neutral fractions, only that of the roots of *B. louisii* showed moderate antitrypanosomal activity against *Tb brucei* in vitro with an IC_50_ value of 20.9 µg/mL.

The isolated compounds and synthetic derivatives were also tested for their antitrypanosomal and antimalarial activities. Compounds **1**–**2** are position isomers (double bond) while **6**–**7** are chain (length) isomers. Both mixtures were showing a single spot on TLC card, indicating their same retardation factor. All the attempts to separate such isomers were unsuccessful independently to the different separation techniques used. Among the tested compounds, the mixtures **1**–**2**, **6**–**7**, and **13** showed good antitrypanosomal activity with IC_50_ values of 4.91, ≤15.2, and 9.50 µM, respectively. From the synthetic derivatives, compound **11b** showed the best antitrypanosomal activity with an IC_50_ value of 8.3 µg/mL while the starting compound, beilschmiedic acid E was inactive against the same strain. Its alcohol derivative (**11a**) exhibited moderate activity. From the results obtained, it appears that the most active compounds are alcohols. The presence of this functional group may be important for the activity. In addition, the activity of the mixtures **1**–**2** and **6**–**7** may be due to synergetic action of their constituents. The antitrypanosomal potential of endiandric acid derivatives is reported here for the first time. The active mixture **6**–**7** consists of two lactones, class of secondary metabolites known whose antitrypanosomal potential was reported previously by Oleivera et al. (2019) [29].

Cytotoxicity assays were also carried out on the extract and compounds, and they were found to be non-toxic. With regard to the antiplasmodial activity, only the synthetic derivative **11b** exhibited a moderate activity with an IC_50_ value of 31.42 µM.

To the best of our knowledge, *B. louisii* has not yet been subject of chemical and biological investigation. The Lauraceae family remains the only source of bioactive endiandric acid derivatives. *Beilschmiedia* and *Endiandra* genus (Lauraceae) are well known for a long time as a rich source of biologically active secondary metabolites. They are still the only source of this class of secondary metabolites [30]. The genus *Beilschmiedia* is a pantropical genus of about 287 species, only 24 species have been studied phytochemically so far. Endiandric acid derivatives are major constituents of these genera, and are important chemotaxonomic markers used to identify *Beilschmiedia* plant species from a phytochemical point of view [3]. In previous studies, endiandric acid has been isolated from *B. obstusifolia*, *B. introrsa*, *B. jonesii*, *B. toorom*, *B. oligandra*, *B. manii*, *B. fulva*, *B. anarcardioides*, *B. erythrophoia*, *B. tsangii*, *B. cryptocaryoides*, and *B. ferruginea*. This is the first time that this class of secondary metabolites is reported from *B. louisii* and *B. obscura*. This study is important as far as the chemotaxonomy of *Beilschmiedia* species is concerned and, and highlights the antitrypanosomal potential of endiandric acid and lactone derivatives.

## 3. Materials and Methods

### 3.1. Chemicals and Reagents

The organic solvents (from Fisher, Waltham, MA, US) used for the extraction of plant material and for column chromatography (CC) were of technical and laboratory grade, while those use in reactions were of analytical grade. For column chromatography, the main solvents used were *n*-hexane, dichloromethane, ethyl acetate, and methanol, used as pure or binary mixtures at different concentrations for purification of compounds. Silica gel 230–400 and 70–230 mesh (Merck) were used for column chromatography with different mixtures of *n*-hexane-ethyl acetate, and dichloromethane-methanol solvent systems as eluents. Iodine or vanillin reagent (1.0 g of vanillin in 70 mL ethanol 96% + 10 mL concentrated sulfuric acid) was used to reveal the spots of compounds on TLC cards.

### 3.2. Instruments and Apparatus

NMR spectra were recorded on Bruker DRX-600 MHz spectrometer (Bruker, Rheinstetten, Germany) operating at 75, 100, 150, 300, 400, and 600 MHz, where chemical shifts (*δ*) are given in ppm with reference to the TMS signal. HR-ESI-MS were recorded on a Bruker Compact Q-TOF mass spectrometer (Bremen, Germany) equipped with DionexUltiMate 3000 UHPLC and electrospray ionization (ESI). For the direct infusion MS, the spectrometer operated in positive and negative modes (mass range: 100–1500, with a scan rate of 1.00 Hz) with automatic gain control to provide high-accuracy mass measurements within 1 ppm deviation using sodium formate as calibrant. The spray voltage was 4.5 kV with a capillary temperature of 200 °C. The flow rate of sample was 180 µL/h and nitrogen were used as sheath gas (4 L/min). Pre-coated aluminum silica gel 60 F_254_ sheets were used for TLC. Spots were visualized with UV light (254 and 365 nm).

### 3.3. Plant Material

The roots and leaves of *B. louisii* were collected in April 2014 at Batshenga, Centre Region of Cameroon. The stem bark of *B. obscura* was collected in April 2016 at Mont Kalla, Centre Region of Cameroon. The plants were identified by Mr Nana Victor, plant taxonomist at the National Herbarium of Cameroon where voucher specimen (*B. obscura*, N° 3360/SRFK and *B. louissii*, N° 6933/SRF/Cam) were deposited.

### 3.4. Extraction and Isolation

The air-dried roots (2.5 kg) and leaves (1.0 kg) of *B. louisii*, and stem bark (2.0 kg) of *B. obscura*, were separately ground and extracted with a mixture of CH_2_Cl_2_-MeOH (1:1) for 48 h at room temperature each (about 25 °C). The different filtrates were freed from solvent under vacuum to give the crude extracts (roots (350.0 g, 14.0%) and leaves (130.0 g, 13.0%) of *B. louisii*, and stem bark (300.0 g, 7.0%) of *B. obscura*). Each extract was partitioned into neutral (roots (300.0 g) and leaves (80.0 g) of *B. louisii*, and stem bark (200.0 g) of *B. obscura*) and alkaloid (roots (8.6 g) and leaves (2.0 g) of *B. louisii*, and stem bark (2.0 g) of *B. obscura*) rich fractions, as described by Pudjastuti [31]. Each crude extract was separately dissolved in cold distilled water and defatted with *n*-hexane. The aqueous layers were then acidified with hydrochloric acid (5% HCl) to pH 3 to obtain acidic solutions that were then extracted with dichloromethane to lead to neutral fractions. The remaining aqueous acidic phases were then made alkaline by adding ammonium hydroxide (25% NH_4_OH) solution to pH 11 and extracted with dichloromethane to afford the alkaloid fractions.

The neutral fraction (180 g) of stem bark of *B. obscura* was subjected to a flash chromatography over silica gel 60 (230–400 mesh, Merck, 300 g) and eluted with mixtures of *n*-hexane/EtOAc and EtOAc/MeOH of increasing polarities. Eighty fractions of 500 mL each were collected and combined according to their TLC profiles into five main sub-fractions (A–E). Sub-fractions A (24.3 g), D (26.8 g), and E (13.3 g) were complex mixtures that were not further studied. Sub-fraction C (52.9 g) was subjected to repeated column chromatography over silica gel (70–230 mesh, Merck, 150 g) and eluted with a mixture of *n*-hexane/EtOAc of increasing polarities (19:1→2:3) to yield the binary mixture of beilschmiedol B and beilschmiedol C (**1** and **2**, 8 mg) and obscurene A (**3**, 10 mg), pentacosan-1-ene (**4**, 35 mg), and 15-phenylpentadecanoic acid (**5**, 15 mg). Sub-fraction B (62.7 g) was equally subjected to column chromatography over silica gel (70–230 mesh, Merck, 230 g) and eluted with the mixture of *n*-hexane/EtOAc of increasing polarities (9:1→7:3) to give the binary mixture of (3*S*,4*R*,5*S*)-4-hydroxy-5-methyl-3-(11′-phenyl-1′-*n*-undecyl)-butanolide (**6**) and (3*S*,4*R*,5*S*)-4-hydroxy-5-methyl-3-(13′–phenyl-1′-*n*-tridecyl)- butanolide (**7**) (**6** and **7**, 30 mg).

The neutral fraction (280 g) of roots *B. louisii* was also subjected to flash chromatography as the previous one to yield sub-fractions (F-L). Fraction G (40 g) was subjected to repeated column chromatography over silica gel (70–230 mesh, Merck, 300 g) and eluted with *n*-hexane and resulted in the isolation of the binary mixture of beilschmiedic acid A and beilschmiedic acid C (**8** and **9**, 80 mg). Fraction F (100 g) was also subjected to column chromatography over silica gel (70–230 mesh, Merck, 200 g) and eluted with a mixture of *n*-hexane/EtOAc (9:1→7:3) to yield compounds beilschmiedic acid D (**10**, 11 mg) and beilschmiedic acid E (**11**, 10 g). Fraction H (17 g) was treated as fraction G to yield methylenedioxyendiandric acid A (**12**, 60 mg) and *β*-sitosterol (**13**, 50 mg). Fraction K (10 g) was purified on a silica gel (70–230 mesh, Merck, 50 g) column with the isocratic mixture of *n*-hexane/EtOAc (1:1) as eluent to yield compounds epicatechin (**14**, 10 mg) and *β*-sitosterol-3-*O*-*β*-D-glucopyranoside (**15**, 75 mg). The neutral fraction of the leaves afforded *β*-sitosterol (**13**, 25 mg), hydrocarbons, and fatty acid compounds, which structures were not determined. Repeated column chromatography of alkaloid fractions of the roots of *B. louisii* and the stem bark of *B. obscura* over alumina gel afforded five compounds that were not characterized due to the non-solubility in NMR solvents.

### 3.5. Semi-Synthetic Reactions

#### 3.5.1. Synthesis of Compound **11a** (Beilschmiedol A)

The solution of Beilschmiedic acid E (**11**) 3 mmol in THF (10 mL) was cooled to −5 °C using ice/NaCl bath and lithium aluminum hydride (LiAlH_4_), 3.5 mmol, was carefully added to the above solution in portions, while maintaining the temperature lower than 0 °C. The mixture was then allowed to rise to room temperature gradually and stirred at room temperature for 48 h. The mixture was quenched using petroleum ether (50 mL) and a solution of 10% of NaOH (30 mL) was added dropwise. The mixture was then extracted with petroleum ether (3 × 50 mL). The organic phases were combined, washed with the saturated brine and dried over MgSO4. The solvent was removed from the resulting solution under vacuum to afford compound **11a** as an off-yellow oil (980 mg, 98%).

#### 3.5.2. Synthesis of Compound **11b** (Beilschmiedal)

In a 100 mL round bottom flask equipped with a magnetic stirrer, a solution of Dess-Martin periodinane (DMP) reagent (3.3 mmol), dissolved in methylene dichloride (DCM, 10 mL) was placed. In a beaker, compound **11a** (3 mmol) was dissolved in 10 mL of DCM, and then slowly added to the DMP solution while stirring. The mixture was stirred at room temperature for 24 h. The reaction mixture was diluted with diethyl ether (50 mL), then quenched by pouring it into a 250 mL beaker containing saturated aqueous sodium bicarbonate solution (15 mL) with sodium thiosulfate (2.5 g). The mixture was stirred for 15 min and transferred into a separation funnel where the layers were allowed to separate. The aqueous layer was discarded, and the organic layer washed with water and saturated brine, successively, and dried over MgSO4. The organic layer was then freed from solvent under vacuum and purified on column chromatography (silica gel; hex-AE (9:1)) to yield compound **11b** as an off-yellow oil (350 mg, 70%).

#### 3.5.3. Synthesis of Compound **11c** (Beilschmiecarbazone)

In a round bottom flask, compound **11b** (20 mmol) was dissolved in 100 mL of ethanol. A solution of thiosemicarbazide reagent (20 mmol dissolved in 20 mL of 1 N hydrochloric acid) was prepared in a beaker and added into a round bottom flask containing the compound **11b**. The mixture was stirred at room temperature (at about 25 °C) to afford compound **11c** (245 mg, 70%) as a precipitate. The precipitate was filtered, dried, and then recrystallized in ethanol (96 °C).

#### 3.5.4. Synthesis of Compound **11d** (Beilschmiegallate)

3,4,5-Trimethoxybenzoic acid (1.0 equiv.) and 4-(*N*,*N*-dimethylamino) pyridine (DMAP, 0.1 equiv.) were dissolved in CH_2_Cl_2_ (10 mL) placed in a round bottom flash and compound **11a** (3.0 equiv.) was added. After 30 min of stirring at room temperature (at about 25 °C), the reaction mixture was cooled to 0 °C and *N*,*N*-dicyclohexylcarbodiimide (DCC; 1.0 equiv.) was added in one portion. The mixture was allowed to stir at room temperature overnight, and urea was filtered off using a kieselguhr plug. The organic solvent was removed under reduced pressure and the organic residue was purified on column chromatography over silica gel using CH_2_Cl_2_ as a solvent to yield compound **11d** (400 mg, 80%).

### 3.6. Spectroscopic Data

Beilschmiedol A (**11a**) yellowish oil. HRESIMS [M + K]^+^ at *m*/*z* 353.2241 (calcd for C_22_H_34_KO, 353.2247), ^13^C–NMR and ^1^HNMR (CDCl_3_), see Table 1.

Beilschmiedal (**11b**) yellowish oil, ^13^C–NMR (CDCl_3_), see Table 2.

Beilschmiecarbazone (**11c**) powder, HRESIMS [M + H]^+^ at *m*/*z* 386.2624 (calcd for C_23_H_34_N_3_S, 386.2630), ^13^C–NMR (CDCl_3_), see Table 2.

Beilschmiegallate (**11d**) colorless oil, HRESIMS [M + K]^+^ at *m*/*z* 547.2820 (calcd for C_32_H_44_KO_5_, 547.2826), ^13^C–NMR (CDCl_3_), see Table 2.

Beilschmiedol B and beilschmiedol C (**1** and **2**) colorless oily, HRESIMS [M + K]^+^ at *m*/*z* 367.2398, ^13^C–NMR and ^1^HNMR (CDCl_3_), see Table 1.

Obscurene (**3**) colorless oil, HRESIMS [M + Na]^+^ at *m*/*z* 239.1770, ^13^C–NMR and ^1^HNMR (CDCl_3_), see Table 3.

Pentacosan-1-ene (**4**) colorless oil, HRESIMS [M − H]^+^ at *m*/*z* 349.1892, ^13^C–NMR and ^1^HNMR (CDCl3), see Table 4.

### 3.7. Biological Activities

#### 3.7.1. In Vitro Antiplasmodial Assay

The antiplasmodial assay was carried out using the pLDH assay as described by Makler. [32] with little modification against the chloroquine-sensitive 3D7 strain of *P. falciparum* (*Pf*3D7). Malaria parasites (*Pf*3D7) were maintained in RPMI 1640 medium containing 2 mM L-glutamine and 25 mM Hepes (Lonza) further supplemented with 5% Albumax II, 20 mM glucose, 0.65 mM hypoxanthine, 60 µg/mL gentamycin, and 2–4% hematocrit human red blood cells. The parasites were cultured at 37 °C under an atmosphere of 5% CO_2_, 5% O_2_, and 90% N_2_ in a sealed T75 culture flask. Single concentration screening was conducted using solutions of 50 µg/mL of the extract and 20 µg/mL of pure or combined compounds. The combinations were obtained by mixing solutions of single corresponding compounds volume by volume. Samples were further added to parasite cultures in 96-well clear plates and incubated in a 37 °C CO_2_ incubator. After 48 h, 20 µL of culture was removed from each well and combined with 125 µL of a mixture of Malstat and NBT/PES solutions in a fresh 96-well plate. These solutions measured the activity of the parasite lactate dehydrogenase (pLDH) enzyme in the cultures. A purple product formed when pLDH was active, was then quantified by Spectramax M3 microplate reader (Abs_620_) from Molecular Devices LLC. The Abs_620_ reading in each well reflected the pLDH activity and hence the number of parasites present. For each sample concentration, % parasite viability–the pLDH activity in sample treated wells relative to untreated controls was calculated. Samples were tested in triplicate and standard deviations (SD) derived. Samples that reduced parasite viability to ≤20% from the single concentration assay were used in a dose response assay to determine the 50% inhibitory concentration (IC_50_) values. For that, samples were serially diluted and tested as described by Makler et al. [32]. Samples were tested in triplicate and for each, the percentage viability was plotted against Log (concentration) and the IC_50_ value obtained from the resulting dose-response curve by non-linear regression. Chloroquine was used as a drug standard and yielded IC_50_ values in the range 0.008 µg/mL.

#### 3.7.2. In Vitro Antitrypanosomal Assay

To assess the trypanocidal activity, samples were added to cultures of *Tb brucei* in 96-well plates at a fixed concentration of 20 μg/mL for single and combined compounds and 50 µg/mL for the extracts. After 48 h of incubation, parasites surviving drug treatment were measured by adding a resazurin. Resazurin is reduced to resorufin (a fluorophore (Exc_560_/Em_590_)) in viable cells and was quantified using a Spectramax M3 microplate reader. Results were expressed as % parasite viability–the resorufin fluorescence in sample treated wells relative to untreated controls. Samples were tested in triplicate and standard deviations (SD) derived. Samples that reduced parasite viability to < 20% were considered for dose-response assay to determine the IC_50_ value. In brief, samples were serially diluted and tested in duplicate. Percentage viability was plotted against Log (concentration) and the IC_50_ obtained from the resulting dose-response curve by non-linear regression. Pentamidine was used as a drug standard and yielded IC_50_ values in the range 0.01 µg/mL [33].

#### 3.7.3. Cytotoxicity Assay

Human cervix adenocarcinoma cells (HeLa cells) were used to check the toxicity of the crude extract and compounds and their combinations as described by Keusch et al. (1972) [34]. Stocks solutions were prepared in DMSO at a concentration of 50 µg/mL for the extracts and 20 µM for compounds, and combinations. To assess the overt cytotoxicity, samples were placed in 96-well plates containing cells for 48 h. The numbers of cells surviving drug treatment were determined using the resazurin based reagent and reading resorufin fluorescence in a Spectramax M3 microplate reader. Results were expressed as % viability–the resorufin fluorescence in compound-treated wells relative to untreated controls. Compounds were tested in duplicate and standard deviations (SD) derived. Emetine (which induces cell apoptosis) was used as the positive control.

## 4. Conclusions

Phytochemical investigation of the MeOH/CH_2_Cl_2_ (1:1) extract of the roots and leaves of *Beilschmiedia louisii* and *B. obscura*, which showed potent antitrypanosomal activity during preliminary screening on *T. brucei* led to the isolation of a mixture of two new endiandric acid derivatives **1** and **2**, and one new phenylalkene (**3**) along with twelve known compounds (**4**–**15**). In addition, four new derivatives (**11a**, **11b**, **11c**, and **11d**) were synthesized from compound **11**. Furthermore, the isolates were evaluated for their antitrypanosomal and antimalarial activities against *Tb brucei* and the *P. falciparum* chloroquine-resistant strain *Pf3D7* in vitro, respectively. From the tested compounds, the mixture of the new compounds exhibited potent antitrypanosomal activity in vitro with IC_50_ value of 1.61 μg/mL. The extracts and compounds from both plants were not toxic against human cervix adenocarcinoma cells (HeLa cells) and the extract of the roots of *B. louisii* exhibited the good antitrypanosomal activity with an IC_50_ value of 4.62 μg/mL.

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
