# Peer review of "Antiparasitic Constituents of Beilschmiedia louisii and Beilschmiedia obscura and Some Semisynthetic Derivatives (Lauraceae)"

_molecules, 2020, doi:10.3390/molecules25122862_

Round 1

Reviewer 1 Report

The manuscript addresses and develops multiple aspects of a topic of great interest.  The research described was well and thoroughly conducted.  Given the many aspects presented (in particular, the chemical and biological ones), the descriptions of the NMR and NS analyses could be clarified so as to render the research accessible to an even larger audience of researchers.  Thus, here are a few suggestions for improving the text:

Page 4 line 74-76

The molecule (6), described as "(2R,3S,4S)-3-hydroxy-4-methyl-2-((11ʹ-phenyl-1ʹ-n-tridecyl) butanolide," should be named "(3S,4R,5S)-4-hydroxy-5-methyl-3-(11’–phenyl-1’-n-undecyl) butanolide."

The molecule (7), described as "(2R,3S,4S)-3-hydroxy-4-methyl-2-(13’ –phenyl-1’-n-undecyl) butanolide," should be named "(3S,4R,5S)-4-hydroxy-5-methyl-3-(13’-phenyl-1’-n-tridecyl) butanolide."

Pages 4-5 line 86-89

“….(Table 1) showed 20 carbon signals…”  In fact, 22 carbon signals are reported subsequently, in lines 87-89.  In addition, the methylene carbon signals reported in both Figs. S1 and S3 should be 7 instead of 9.

Page 11 Fig 1

The diagram of the alkyl chain of the molecule 11b (group R3) should be the same as R3 in 11c. 

Page 14 Paragraph 2.2 In vitro antiplasmodial......

A sentence could be inserted to highlight the synergistic effect that, at least apparently, has the total extract of B. louisii (roots and leaves) with respect to the neutral fraction and alkaloid fraction.

Author Response

Pages

Lines

Comments of the Reviewer

Authors  answers to Reviewer

4

74-76

The molecule (6), described as "(2R,3S,4S)-3-hydroxy-4-methyl-2-((11ʹ-phenyl-1ʹ-n-tridecyl) butanolide," should be named "(3S,4R,5S)-4-hydroxy-5-methyl-3-(11’–phenyl-1’-n-undecyl) butanolide."

The molecule (7), described as "(2R,3S,4S)-3-hydroxy-4-methyl-2-(13’ –phenyl-1’-n-undecyl) butanolide," should be named "(3S,4R,5S)-4-hydroxy-5-methyl-3-(13’-phenyl-1’-n-tridecyl) butanolide."

The molecule (6) has been renamed as "(3S,4R,5S)-4-hydroxy-5-methyl-3-(11ʹ-phenyl-1ʹ-n-undecyl)-butanolide," as proposed by the reviewer.

For the same reasons compound (7) was described as "(3S,4R,5S)-4-hydroxy-5-methyl-3-(13ʹ –phenyl-1ʹ-n-tridecyl)- butanolide," as suggested by the reviewer.

4-5

86-89

 “….(Table 1) showed 20 carbon signals…”  In fact, 22 carbon signals are reported subsequently, in lines 87-89.  In addition, the methylene carbon signals reported in both Figs. S1 and S3 should be 7 instead of 9.

Corrected as “….(Table 1) showed 22 carbon signals…”  accordingly, with the 22 carbon signals reported in lines 87-89.  

On Figs. S1 and S3 we can see that the signal at 29.8 is broad than others, suggesting that signals are overlapped. In addition, the low resolution mass spectra was in accord with 22 Carbons

11

Fig 1

The diagram of the alkyl chain of the molecule 11b (group R3) should be the same as R3 in 11c. 

Fig 1

The diagram of the alkyl chain R3 of the molecule 11b has been changed as in 11c as shown in red on Fig. 1.

14

Paragraph

2.2 In vitro antiplasmodial......

A sentence could be inserted to highlight the synergistic effect that, at least apparently, has the total extract of B. louisii (roots and leaves) with respect to the neutral fraction and alkaloid fraction.

The non-alkaloid fraction of the roots extract and the total extract of leaves of B. louissi was moderately active, while the alkaloid fraction was non-active.

Regarding the leaves, only the total extract showed a moderate activity. In both cases, the synergetic effect of the alkaloid and the non-alkaloid fractions could be evident.

Reviewer 2 Report

Compound1 3 position: The chemical shift is too height. This chemical shift is almost same as 3 position of compound 2, which would not be allylic position.

The solvent, temperature, reaction time of oxidation and esterification should be shown in figure 1.

The yield of Dess-Martin oxidation is 70%. The staring material was recovered?

Table 5.  The μmol/L should be employed.  The more discussion on the biological activity is required, for example effect of structure, comparison with previous report.

Author Response

Pages

Lines

Comments of the Reviewer

Authors  answers to Reviewer

Compound 1 3 position: The chemical shift is too height. This chemical shift is almost same as 3 position of compound 2, which would not be allylic position.

The attribution of the chemical shift of C-3 in compound 1 was deduced from that of compound 11a where correlations were observed between olefinic protons (H4 and H5) and C-3. This was further confirmed by observing other endiandric acids with the same skeleton and nearly substitution.

Ping‐Shin Yang, Ming‐Jen Cheng, Jih‐Jung Chen, Ih‐Sheng Chen. Two New Endiandric Acid Analogs, a New Benzopyran, and a New Benzenoid from the Root of Beilschmiedia erythrophloia. Helvetica 2008, 91(11), 2130-2138.

Mohamad Nurul Azmi, Charlotte Gény, Aurélie Leverrier, Marc Litaudon, Vincent Dumontet, Nicolas Birlirakis, Françoise Guéritte, Kok Hoong Leong, Siti Nadiah,  Halim, Khalit Mohamad, Khalijah Awang.  Kingianic Acids A–G, Endiandric Acid Analogues from Endiandra kingiana. Molecules 2014, 19(2), 1732-1747; https://doi.org/10.3390/molecules19021732

The solvent, temperature, reaction time of oxidation and esterification should be shown in figure 1.

The solvent, temperature, reaction time of oxidation and esterification are shown in red in Figure 2.

The yield of Dess-Martin oxidation is 70%. The staring material was recovered?

We stopped the reaction when the reaction was almost gone. During the separation by chromatography we were able to recover a very small quantity of the starting material.

Table 5.  The μmol/L should be employed.  The more discussion on the biological activity is required, for example effect of structure, comparison with previous report.

Table 5

The results were expressed in μg/mL for the biological activities of extracts and compounds for homogeneity. Since the biological activity of some mixtures have been evaluated as well.

Reviewer 3 Report

The manuscript describes in vitro antiparasitic activity of compounds isolated from Beilschmiedia louisii and Beilschmiedia obscura and their semisynthetic derivatives against Plasmodium falciparum and Trypanosoma brucei brucei. I recommend it for publication after certain revisions.

Below there are some specific comments and suggestions:

Since abstract is independent part of the paper, the numbers of compounds should be replaced by their chemical names.

Page 16, lines 259-261: It should be explained why compounds 1-2 and 6-7 have been tested as their mixtures.

Page 16, lines 266-267: Are there any data supporting hypothesis about synergistic action of tested compounds?

Page 16, lines 271-274: Two individual sentences should not be presented as two separated paragraphs.

Line 17, lines 282-283: Contribution of the study to clarification of chemotaxonomical relationship of species within the genus Beilschmiedia should be explained.

Style of citations should be checked carefully. For example, “Pratiwi et al., 2010 [29]” should be corrected to “Pratiwi et al. [29]” (page 18, line 322).

Pages 22 and 23, lines 418 and 438: Delete repeating citation “Makler et al. [30]”.

In the Materials and Methods section, the provider/producer of equipment used in experiments should clearly be specified (e.g. for Spectramax M3 microplate reader, page 23, lines 431-432).

Page 25, lines 473 and 474: Both sentences are starting with expression “In addition”. Modify it.

Author Response

Pages

Lines

Comments of the Reviewer

Authors  answers to Reviewer

2

28-29

Since abstract is independent part of the paper, the numbers of compounds should be replaced by their chemical names.

Only the names of new derivatives were added. If we add the names of all isolates instead of their numberings, we will exceed the allowed 200 words for the abstract as per the journal.

16

259-261

 It should be explained why compounds 1-2 and 6-7 have been tested as their mixtures.

Compounds 1-2 are position isomers (double bond) while 6-7 are chain (length) isomers. Both mixtures were showing a single spot on TLC card, indicating their same retardation factor. All the attempts to separate such isomers were unsuccessful independently to the different separation techniques used.

16

266-267

 Are there any data supporting hypothesis about synergistic action of tested compounds?

Not yet. In fact, compounds 1 and 2 are reported here for the first time. So, it could not be possible to have such hypothesis existing.

16

271-274

Two individual sentences should not be presented as two separated paragraphs.

Both sentences has been combined to form a paragraph.

17

282-283

Contribution of the study to clarification of chemotaxonomical relationship of species within the genus Beilschmiedia should be explained.

The Lauraceae family remains the only source of bioactive endiandric acid derivatives. Beilschmiedia and Endiandra genus (Lauraceae) are well known for a long time as a rich source of biologically active secondary metabolites. They are still the only source of this class of secondary metabolites (Lenta et al., 2015). The genus Beilschmiedia is a pantropical genus of about 287 species, only 24 species have been studied phytochemically so far. Endiandric acid derivatives are major constituents of these genus, and are important chemotaxonomic markers used to identify Beilschmiedia plant species from a phytochemical point of view (Wan et al., 2015).

Lenta N.B.; Chouna J.R.; Nkeng-Efouet P.A.; Sewald N. Endiandric acid derivatives and other constituents of plants from the Genera Beilschmiedia and Endiandra (Lauraceae). Biomolecules 2015, 5, 910-942.

Wan M.N.H.W.S.; Wan S.; Farediah A.; Khong H.Y.; Razauden M.Z. A Review on Chemical Constituents and Biological Activities of the Genus Beilschmiedia (Lauraceae). Trop. J. Pharm. Res. 2015, 14 (11), 2139-2150.

18

322

Style of citations should be checked carefully. For example, “Pratiwi et al., 2010 [29]” should be corrected to “Pratiwi et al. [29]”

The reference has been corrected as Pudjastuti et al. [32].

22-23

418-438

Delete repeating citation “Makler et al. [30]”.

The repeated reference has been deleted and the remaining one has been corrected as Makler et al. [33]

23

431-432

In the Materials and Methods section, the provider/producer of equipment used in experiments should clearly be specified (e.g. for Spectramax M3 microplate reader, page, lines).

It has been corrected in the text as follow “was then quantified by Spectramax M3 microplate reader (Abs620) from Molecular Devices LLC.”

25

473-474

Both sentences are starting with expression “In addition”. Modify it.

Corrected as

“In addition, four new derivatives (11a, 11b, 11c, and 11d) were synthesized from compound 11. Furthermore, the isolates were evaluated for their antitrypanosomal and antimalarial activities against Tb brucei and the P. falciparum chloroquine-resistant strain Pf3D7 in vitro, respectively.”

Round 2

Reviewer 2 Report

Table 5.  The μmol/L should be employed.  We can not evaluate the different compounds with different molecular weight withμg/mL.

Author Response

Table 5.  The μmol/L (μM) has been employed for pure compounds has requested by the reviewer.  IC50 are then expresses in μg/mL for extracts, fractions and in µM for isolated pure compounds; * IC50 maximum was calculated with the constituent having the lowest molecular mass in the mixture (mixture of compounds 6 and 7, chain isomers)